# AUGMENTED POLICY GRADIENT METHODS FOR EFFICIENT REINFORCEMENT LEARNING

## ABSTRACT

We propose a new mixture of model-based and model-free reinforcement learning (RL) algorithms that combines the strengths of both RL methods. Our goal is to reduce the sample complexity of model-free approaches utilizing fictitious trajectory rollouts performed on a learned dynamics model to improve the data efficiency of policy gradient methods while maintaining the same asymptotic behaviour. We suggest to use a special type of uncertainty quantification by a stochastic dynamics model in which the next state prediction is randomly drawn from the distribution predicted by the dynamics model. As a result, the negative effect of exploiting erroneously optimistic regions in the dynamics model is addressed by next state predictions based on an uncertainty aware ensemble of dynamics models. The influence of the ensemble of dynamics models on the policy update is controlled by adjusting the number of virtually performed rollouts in the next iteration according to the ratio of the real and virtual total reward. Our approach, which we call Model-Based Policy Gradient Enrichment (MBPGE), is tested on a collection of benchmark tests including simulated robotic locomotion. We compare our approach to plain model-free algorithms and a model-based one. Our evaluation shows that MBPGE leads to higher learning rates in an early training stage and an improved asymptotic behaviour.

## 1 INTRODUCTION

Reinforcement Learning (RL) can be broadly classified in two categories: model-free RL, which uses experience from interacting with the environment to learn a value function, and model-based RL, which uses experience to approximate the environment by a model (Deisenroth & Rasmussen (2011)). Recently, impressive results were achieved applying model-free RL aproaches to challenging tasks. A RL agent with a neural network (NN) based Q-value representation was introduced in Mnih et al. (2015) which was able to reach an equivalent or higher performance on many Atari games compared to humans by learning a state function based on raw pixels. Further modifications addressed the problem of Q-value overestimation (van Hasselt et al. (2015)), dropping significant experience (Schaul et al. (2015)) and extended the approach to asynchronous methods (Mnih et al. (2016)) to improve the overall performance. Beside learning a value function, it is possible to use the gradient of the RL objective to update the policy (Peters & Schaal (2006), Sutton & Barto (2010), Silver et al. (2014), Lillicrap et al. (2015)). Actor-critic methods aim at combining the strong points of policy gradients and an approximated value function, in which the critic learns an approximated value function that is used to update the policy of the actor in the direction of its gradient (Konda & Tsitsiklis (2000), Schulman et al. (2015a), Mnih et al. (2016), Gu et al. (2016)).

In contrast, model-based RL can be significantly more data efficient. It is notable that a dynamics model is not necessarily globally valid. In guided policy search (GPS), time-varying linear dynamics models are learned from multiple rollouts with diverse start and goal conditions (Levine & Koltun (2013a), Levine & Koltun (2013b), Levine & Koltun (2014)). These trajectory samples are then used to fit a time-varying Gaussian dynamics model leading to a Gaussian distribution over trajectories, even for unknown dynamics (Levine & Abbeel (2014)). The neural network policy is trained in a supervised fashion such that local policies are improved minimizing a cost function bounded by the trust region of the KL-divergence, and the weights of the NN policy are adjusted to mimic the trajectory distribution. Further improvements extended GPS to a broader robust

state space by introducing a new policy representation which is called general motor reflex (GMR) (Ennen et al. (2018)). In a first step, the state space is encoded to a latent state representation by a variational autoencoder. Then, a translation model predicts motor reflex parameters in order to adjust a Gaussian controller resulting in robust trajectories even outside the distribution of training samples.

However, modelling errors can reduce the efficiency and benefit of model-based RL algorithms since model imperfections are exploited (Deisenroth & Rasmussen (2011), Schneider (1996), Atkeson & Santamaria (1997)) known as model-bias. With model-bias being a result of overfitting, one approach to solve this problem is to incorporate explicit uncertainty estimates by using a probabilistic dynamics model. Probabilistic inference for learning control (PILCO) (Deisenroth & Rasmussen (2011)) uses a Gaussian process (GP) to represent the dynamics of the environment. Such a model provides predictions with true Bayesian uncertainty estimates but suffer from the curse of dimensionality (Calandra et al. (2014)). Recently, Bayesian neural networks (BNN) received much attention. Approximate Bayesian techniques, e.g. variational inference (Graves (2011)) or Monte-Carlo dropout (Gal & Ghahramani (2015)), come at the drawback that correlations between parameters are not maintained, while scaling abilities of the Bayesian 'gold standard' inference method Markov-Chain-Monte-Carlo (MCMC) are limited (Hinton & Neal (1995). Adding a regularization term to the loss function of a NN, e.g. Tikhonov regularization, permits maximum a posterior (MAP) estimates of the parameters since the solution of the regularized loss function is equivalent to the maximum of the posterior density function (Bardsley (2012)). However, the regularized solution leads to a point estimator of the true posterior density. This insufficiency can be addressed when sampling from the true posterior distribution using a MCMC method providing additional information about the shape of the posterior but is computationally more demanding.

Our approach is inspired by recent advances, Model-Ensemble Trust-Region Policy Optimization (ME-TRPO) (Kurutach et al. (2018)), Model Based Meta Policy Optimization (MB-MPO) (Clavera et al. (2018)), Model-assisted Boostrapped DDPG (MA-BDDPG) (Kalweit & Boedecker (2017)), and Model-Based Value Expansion (MVE) (Feinberg et al. (2018)). In ME-TRPO, a set of samples from the real environment is used to train an ensemble of NNs. In a second iteration, fictitious trajectory rollouts are generated to update the policy using Trust Region Policy Optimization (TRPO) (Schulman et al. (2015a)), i.e., the policy update is only based on samples from the approximated dynamics model of the environment. In MB-MPO, the meta policy learning framework 'model agnostic meta learning' (Finn et al. (2017)) is used to train a policy on an ensemble of learned dynamics models. The notion is that the meta-learning frame of the dynamics models is used to adapt quickly to any new dynamics model within one update step. The overall similarities of all dynamics models are incorporated in the meta-model that is then adjusted to create the fitted dynamics model. MA-BDDPG employs the same idea to augment the dataset to achieve a better data efficiency. In contrast to our approach, MA-BDDPG makes use of an actor-critic method in which the policy is represented deterministically. Updates in the actor and critic network follow the advances of Deep Q-Networks by refering to time-delayed target networks. A further difference is in the procedure for estimating uncertainty, i.e. Kalweit & Boedecker (2017) applies bootstrapping (Efron & Tibshirani (1998)) to get a distribution over Q-functions while we use an anchored ensemble of NNs. MVE improves value estimates assuming that an approximated dynamics model can be used in training a critic up to a depth $H$ in which we are certain about the model accuracy. Contrary to our approach, MVE defines its dynamics model as a deterministic neural network.

Our new algorithm MBPGE is a model-free RL algorithm that is augmented by a true Bayesian ensemble of dynamics models to generate fictitious trajectory rollouts to decrease the sample complexity. In contrast to ME-TRPO, we take advantage of a very recent inference advance based on randomised anchored MAP sampling that allows for a true Bayesian uncertainty estimate of the true posterior (Pearce et al. (2018)). As a result, our stochastic dynamics model reduces the effect of policy overfitting as model inaccuracies will be addressed by highly random next state predictions. We have seen that ill-distributed datasets, i.e., data points of some regions in the state space are overrepresented while others are underrepresented, might lead to enormous fictitious total rewards that diverge heavily from those of the real environment samples due to overfitting dynamics models. We tackle this issue by introducing a ratio of fictitious and real rewards so that our algorithm

adjusts naturally to become more model-free if the dynamics model is inaccurate and acts in a more model-based way if the approximation of the environment is accurate. To summarize, our main contributions are:

1. A novel model-free RL algorithm that uses a true Bayesian representation for the stochastic dynamics model and

2. a routine that adjusts the composition of the dataset for a policy gradient estimate by weighting the trajectory samples based on the ratio of the total rewards of the real environment and the approximated model.

We derive our approach by reviewing the background in constrained policy gradient methods and randomized maximum a posterior sampling. Then, we present our augmented model-free reinforcement learning algorithm in detail. Finally, we evaluate our approach on challenging simulated tasks in a continuous control domain.

## 2 Preliminaries

We formulate the underlying problem of our tasks as a discrete-time finite-horizon Markov decision process (MDP), defined by the tuple $(\mathcal{S}, \mathcal{A}, P, r, \rho_0, \gamma, T)$. Here, we assume that our agent acts in a stochastic environment with continuous state space $\mathcal{S} \subseteq \mathbb{R}^n$ and continuous action space $\mathcal{A} \subseteq \mathbb{R}^m$. Additionally, we assume that an unknown dynamics model $P(\boldsymbol{s}_{t+1}|\boldsymbol{s}_t, \boldsymbol{a})$ is defined for our environment and we are given a reward function $r(\boldsymbol{s}_t, \boldsymbol{a}_t)$. Let $\rho_0$ denote the initial state distribution, $\gamma$ is the discount factor, $T$ is the finite horizon and $\pi_\theta(\boldsymbol{a}|\boldsymbol{s})$ denotes our neural network policy, parameterized by $\boldsymbol{\theta}$, distributed over actions $\boldsymbol{a}$, and conditioned to states $\boldsymbol{s}$. The overall aim is to learn an optimal policy $\pi$ such that the expected total reward $J(\pi) = \mathbb{E}_{\tau \sim \pi} \left[ \sum_{t=0}^T rs(s_t, a_t) \right]$ is maximized, where $\tau = s_0, a_0, ..., a_{T-1}, s_T$ denotes a trajectory with $s_0 \sim \rho_0$, $a_t \sim \pi_\theta$ and $s_{t+1} = P(s_t, a_t)$.

### 2.1 Policy gradient methods

The maximization problem of the RL objective can be solved by performing gradient ascent computing the derivative with respect to the policy parameters $\theta$. The major problem of policy gradient methods is to find a good estimator for the gradient since the variance of the gradient estimator increases with the time horizon. Fortunately, a variance reduction scheme for policy gradients, generalized advantage estimator (GAE) (Schulman et al. (2015b)), reduces the variance significantly while only introducing a small bias. It is possible to trade-off between bias and variance by using a $\lambda-$return (Sutton & Barto (2010)). Extending the idea of an n-step return, one can weight each n-step update by $\lambda^{n-1}$ and normalize the sum by $(1 - \lambda)$ resulting in

$$R_t^\lambda = (1 - \lambda) \sum_{n=1}^\infty \lambda^{n-1} R_t^n \ . \tag{1}$$

The finite horizon version assumes that all rewards after time step $T$ equal 0, leading to

$$R_t^\lambda = (1 - \lambda) \sum_{n=1}^{T-t-1} \lambda^{n-1} R_t^n + \lambda^{T-t-1} R_t^{T-t} \ . \tag{2}$$

Setting $\lambda = 0$ results in the biased single step return $R_t^{(1)}$ with low variance, while $\lambda = 1$ corresponds to the unbiased Monte-Carlo return $R_t^{(\infty)}$ with high variance. GAE can be derived when estimating the advantage $A^\pi(s_t, a_t) = Q^\pi(s_t, a_t) - V^\pi(s_t)$ with the $\lambda-$return. The gradient of the objective now turns to

$$g := \nabla_\theta \mathbb{E}\left[\sum_{t=0}^T r_t\right] = \mathbb{E}\left[\sum_{t=0}^T \hat{A}_t \nabla_\theta log\pi_\theta(a_t|s_t)\right] \ . \tag{3}$$

Usually, we intend to update our parameters only by small changes. However, instead of performing update steps in parameter space using the Euclidean distance, it is much more convenient to

measure closeness between the current policy and the updated policy in terms of distances between distributions. The intuition behind is that distances between distributions, e.g. Kullback-Leibler divergence (Kullback & Leibler (1951)) or Hellinger distance (Hellinger (1909)), are invariant to scaling or shifting. This type of approach is known as natural policy gradients (Kakade (2001), Sutton et al. (2000), Peters & Schaal (2006)). Trust region policy optimization (TRPO) utilizes this idea and maximizes its 'surrogate' objective function by constraining the allowed policy update step (Schulman et al. (2015a))

$$\max_\theta \hat{\mathbb{E}}_t \left[ \frac{\pi_\theta(a_t|s_t)}{\pi_{\theta_{old}}(a_t|s_t)} \hat{A}_t \right] \quad s.t. \quad \hat{\mathbb{E}}_t \left[ KL \left[ \pi_{\theta_{old}}(*|s_t), \pi_\theta(*|s_t) \right] \right] \leqq \delta \ . \tag{4}$$

TRPO builds on the concept of guaranteed monotonic improvement. The notion is that we can compute an upper bound for the error of diverging policies. Therefore, we can guarantee a policy improvement as long as we optimize the local approximation within a trusted region. The proof is given in Schulman et al. (2015a). PPO combines the appealing guarantee of improvement with a much simpler implementation (Schulman et al. (2017)). Instead of maximizing the 'surrogate' function in eq. 4, it uses the following clipped objective

$$J(\theta) = \hat{\mathbb{E}}_t \left[ min(r_t(\theta)\hat{A}_t, clip(r_t(\theta), 1 - \epsilon, 1 + \epsilon)\hat{A}_t \right] \ . \tag{5}$$

Clipping the probability ensures that the probability ratio $r_t(\theta) = \frac{\pi_\theta(a_t|s_t)}{\pi_{\theta_{old}}(a_t|s_t)}$ is in the interval $[1 - \epsilon, 1 + \epsilon]$, while taking the minimum of the clipped and unclipped objective leads to a lower bound of the objective with the same convenient effect as constraining the 'surrogate' objective in TRPO.

## 2.2 MODEL-BASED REINFORCEMENT LEARNING WITH AN ANCHORED ENSEMBLE OF DEEP NEURAL NETWORKS

Contrary to model-free RL, model-based RL utilizes the interactions with the environment to learn an approximated model. Model-based approaches benefit from a significantly lower sample complexity. However, learning an accurate model of the environment can be very challenging for certain domains. As a consequence, model-based algorithms often show a worse asymptotic behaviour. To date, there are several diverse approaches to represent the dynamics model and a proper choice is often crucial. Dynamics model in RL must reliably fit to datasets with a low and high number of data points: during the first few iterations only a limited number of data points is available, causing expressive general function approximators, e.g. deep neural networks, to overfit. With progressing iterations simple function approximators tend to underfit complex system dynamics.

Bayesian regression aims to find the parameters $\theta$ of a function $y* = f_\theta(x*)$, which are likely to have generated the output $Y$ given input $X$. This can be achieved by updating a prior distribution over the parameters of the neural network $\theta_0 \sim p(\theta)$ by using Bayes Theorem. If it is possible to compute the posterior distribution

$$p(\theta|X,Y) = \frac{p(Y|X,\theta)p(\theta)}{p(Y|X)} \ , \tag{6}$$

the posterior distribution of new data points $y*$ can be inferred by marginalizing over $\theta$

$$p(y*|x*,X,Y) = \int (Y|X,\theta)p(\theta)d\theta \ . \tag{7}$$

Previous work revealed that exploiting uncertainty estimation in RL can increase the sample efficiency dramatically (Deisenroth & Rasmussen (2011)). Gaussian Processes (Rasmussen & Williams (2008)) perform well in low data regimes but do not scale to complex tasks with high dimensionality (Calandra et al. (2014)). On the other hand, probabilistic NNs can approximate arbitrarily complex dynamics but come at the cost that a true Bayesian uncertainty estimate cannot be provided (Gal & Ghahramani (2015)). The computation of the integral in eq. 7 can in general only be approximated numerically under a trade-off between accuracy and performance.

A Baysian neural network (BNN) can be approximated by using a diverse ensemble $F = \{f_1, ..., f_K\}$ of NNs. Ensembling provides an uncertainty estimate in such way that the variance

of the ensemble's prediction is considered to be its uncertainty. Technically, this approach is not Bayesian (Gal (2016)). This drawback was addressed in Pearce et al. (2018) by regularizing the parameters about values drawn from a prior distribution which is called randomized anchored maximum a posterior sampling. This approach uses Bayes' rule to compute a multivariate posterior by

$$\mathcal{N}(\boldsymbol{\mu}_{post}, \Sigma_{post}) \propto \mathcal{N}(\boldsymbol{\mu}_{prior}, \Sigma_{prior})\mathcal{N}(\boldsymbol{\mu}_{like}, \Sigma_{like}) \tag{8}$$

with $\mathcal{N}(\boldsymbol{\mu}_{prior}, \Sigma_{prior})$ as multivariate prior and $\mathcal{N}(\boldsymbol{\mu}_{like}, \Sigma_{like})$ as multivariate likelihood. When following the MAP approach $\mu_{MAP} = \mu_{post}$, we can use a parameterized mean of the prior $\mu_{prior,\theta_0}$ so that the true posterior distribution is matched with

$$\boldsymbol{\mu}_{MAP}(\boldsymbol{\theta}_0) = (\boldsymbol{\Sigma}_{like}^{-1} + \boldsymbol{\Sigma}_{prior}^{-1})^{-1}(\boldsymbol{\Sigma}_{like}^{-1}\boldsymbol{\mu}_{like} + \boldsymbol{\Sigma}_{prior}^{-1}\boldsymbol{\mu}_{prior,\theta_0}) \,. \tag{9}$$

According to Pearce et al. (2018) suitable parameters for $\boldsymbol{\theta}_0$ can be found by setting $\mathbb{E}\left[\boldsymbol{\mu}_{MAP}(\boldsymbol{\theta}_0)\right] = \boldsymbol{\mu}_{post}$ and $\mathbb{V}\left[\boldsymbol{\mu}_{MAP}(\boldsymbol{\theta}_0)\right] = \boldsymbol{\Sigma}_{post}$ resulting in $\boldsymbol{\theta}_0 \sim \mathcal{N}(\mu_0, \Sigma_0)$ with

$$\boldsymbol{\mu}_0 = \boldsymbol{\mu}_{prior} \tag{10}$$

$$\boldsymbol{\Sigma}_0 = \boldsymbol{\Sigma}_{prior} + \boldsymbol{\Sigma}_{prior}^2\boldsymbol{\Sigma}_{like}^{-1} \approx \boldsymbol{\Sigma}_{prior} \,. \tag{11}$$

When using NNs in an anchored ensemble, the loss function needs to be modified. The typical loss function in a neural network is defined as

$$L = \frac{1}{N}||\boldsymbol{y} - \hat{\boldsymbol{y}}||_2^2 + \frac{1}{N}||\boldsymbol{\Gamma}^{1/2}\boldsymbol{\theta}||_2^2 \tag{12}$$

where $\hat{\boldsymbol{y}}$ denotes the predictions of the neural net, $\boldsymbol{y}$ is the vector of the corresponding labels, $\boldsymbol{\theta}$ is the flattened vector of NN parameters, and $\boldsymbol{\Gamma}$ is the diagonal square matrix of a $L_2$ regularization. Minimizing the loss in eq. 12 results in parameters that are equal to MAP estimates with a zero mean normal prior. Eq. 12 can be modified in such way that the minimized parameters can be seen as MAP estimates with non-zero centered priors

$$L_{anchored} = \frac{1}{N}||\boldsymbol{y} - \hat{\boldsymbol{y}}||_2^2 + \frac{1}{N}||\boldsymbol{\Gamma}^{1/2}(\boldsymbol{\theta} - \boldsymbol{\theta}_0)||_2^2 \,. \tag{13}$$

The proof is provided in Pearce et al. (2018). As a result, randomised anchored MAP sampling with NN leads to a good approximation of the true posterior. The algorithm for training an anchored ensemble with deep neural networks is given in Algorithm 1. In the initialization, an ensemble of $K$ neural networks with different parameters $\boldsymbol{\theta}_{j,0}$ is created. The parameters are drawn from a distribution over the network weights $\boldsymbol{\theta}_{j,0} \sim \mathcal{N}(\boldsymbol{\mu}_0, \boldsymbol{\Sigma}_0)$ with $\boldsymbol{\mu}_0$ according to eq. 10 and $\boldsymbol{\Sigma}_0$ as presented in eq. 11 (see Alg. 1, 3-8). During training, every individual neural network is trained with the training data $\mathbf{X}$ and the corresponding labels $\mathbf{Y}$. The loss is computed according to eq. 13 (see Alg. 1, 9-11). Mean and variance of the true posterior distribution are estimated for the ensemble by

$$\hat{\boldsymbol{y}}(\mathbf{x}*) = \frac{1}{K}\sum_{j=1}^{K} f_j(\mathbf{x}*) \tag{14}$$

$$\hat{\boldsymbol{\sigma}}_y^2(\mathbf{x}*) = I\hat{\sigma}_\epsilon^2 + \frac{1}{K}\sum_{j=1}^{K}(f_j(\mathbf{x}*) - \hat{\boldsymbol{y}})^T(f_j(\mathbf{x}*) - \hat{\boldsymbol{y}}) \,, \tag{15}$$

where $f_j$ denotes an individual NN within the set of $K$ NNs of the ensemble, $\hat{\sigma}_\epsilon^2$ is an estimate of the noise of the training data, $\hat{\boldsymbol{y}}(\boldsymbol{x}*)$ is the predicted mean of the ensemble and $\hat{\boldsymbol{\sigma}}_y^2$ denotes the predicted variance of the ensemble at test point $\boldsymbol{x}*$.

We make use of the uncertainty estimate in eq. 15 by sampling the next state prediction from the distribution predicted by the approximated BNN, i.e. $s_{t+1} \sim \mathcal{N}(\hat{\boldsymbol{y}}(s_t, a_t), \hat{\boldsymbol{\sigma}}_y^2(s_t, a_t))$. Consequently, the rollouts sampled from the learned dynamics model will diverge much in regions of high uncertainty which prevents the policy to overfit to optimistic model inaccuracies. The efficiency of this approach was empirically proven by Kurutach et al. (2018).

---

**Algorithm 1** Randomized MAP sampling with anchored ensembles

---
1: **input:** Training data $\mathbf{X}$ & $\mathbf{Y}$, vector test data point $\mathbf{x}*$, prior mean $\boldsymbol{\mu}_{prior}$ and prior covariance $\boldsymbol{\Sigma}_{prior}$, ensemble size $K$, data noise variance estimate $\hat{\sigma}_{\epsilon}^{2}$
2: **output:** estimate of mean $\hat{\boldsymbol{y}}$ and variance $\hat{\boldsymbol{\sigma}}_{y}^{2}$
    *# Ensemble Initialization*:
3: $\boldsymbol{\Gamma} = \hat{\sigma}_{\epsilon}^{2}\boldsymbol{\Sigma}_{prior}^{-1}$
4: **for** $j = 1$ **to** $K$ **do**
5:     $\boldsymbol{\mu}_{0} = \mu_{prior}$ (eq. 10), $\Sigma_{0} = \Sigma_{prior}$ (eq. 11)
6:     sample $\boldsymbol{\theta}_{j,0}$ from $\mathcal{N}(\boldsymbol{\mu}_{0}, \Sigma_{0})$
7:     create neural network $f_{j}$ with $\boldsymbol{\Gamma}, \boldsymbol{\theta}_{j,0}$
8: **end for**
    *# Ensemble Training:*
9: **for** $j = 1$ **to** $K$ **do**
10:     train $f_{j}$ with $\mathbf{X}, \mathbf{Y}$, loss in eq. 13
11: **end for**
    *# Ensemble Prediction:*
12: $\hat{\boldsymbol{y}}(\mathbf{x}*) = \frac{1}{K}\sum_{j=1}^{K} f_{j}(\mathbf{x}*)$ (eq. 14)
13: $\hat{\boldsymbol{\sigma}}_{y}^{2}(\mathbf{x}*) = I\hat{\sigma}_{\epsilon}^{2} + \frac{1}{K}\sum_{j=1}^{K}(f_{j}(\mathbf{x}*) - \hat{\boldsymbol{y}})^{T}(f_{j}(\mathbf{x}*) - \hat{\boldsymbol{y}})$ (eq. 15)

---

# 3 AUGMENTED MODEL-FREE REINFORCEMENT LEARNING

In this section, we describe our gradient based model-free RL algorithm that uses BNNs with randomized MAP sampling as explained in the previous section. Our motivation is to address complex and high-dimensional real robotic tasks in continuous action and state spaces. This aim requires model-free RL algorithms to become more data efficient while maintaining the same asymptotic behaviour. Our approach, model-based policy gradient enrichment, achieves this goal by amending model-free RL with an ensemble of deep NN based dynamics models. Using randomized anchored MAP sampling our approach does not only compute a point estimate of the true posterior distribution but also accounts for the shape of the posterior distribution (Pearce et al. (2018)). This does not only provide a true Bayesian uncertainty estimate, but also prevents overfitting in the approximated stochastic dynamics model.

## 3.1 THE MODEL-FREE REINFORCEMENT LEARNING FRAME

The first step of our proposed algorithm follows the commonly accepted procedure of model-free RL algorithms in which we iteratively perform trajectory rollouts to collect data and estimate the gradient (see Alg. 2, 6-7). Policy improvement is performed following the clipped 'surrogate' objective shown in eq. 5 (see Alg. 2, 12). Additionally, we use an advantage estimator (Schulman et al. (2015b)) to reduce variance when estimating the policy gradient. However, we do not discard the collected samples of the real environment when performing a policy update. Instead, we add the samples to our dataset $\mathcal{D}$ in order to train our stochastic dynamics model. In MBPGE, the batch of trajectories $\mathcal{U}_{\mathcal{PPO}}$ needed for each unique policy update is sampled partially from the real environment (see Alg. 2, 7) and partially from the approximated dynamics model (see Alg. 2, 9). First, the trajectories $\tau$ will be collected from the real environment and added to $\mathcal{U}_{\mathcal{PPO}}$ and $\mathcal{D}$. In a second step, the anchored ensemble of NNs is trained based on all trajectory samples of the real environment (see Alg. 2, 8). The complete algorithm is given in Algorithm 1. This ensemble is used to collect fictitious trajectories $\tilde{\tau}$ which are added to $\mathcal{U}_{\mathcal{PPO}}$. Afterwards, a standard policy update according to proximal policy optimization (PPO) (Schulman et al. (2017)) based on the batch $\mathcal{U}_{\mathcal{PPO}}$ will be performed. Depending on the composition of sampled trajectories in the sample batch $\mathcal{U}_{\mathcal{PPO}}$, we can effect the influence of the stochastic dynamics model on the policy update. If the learned dynamics model is inaccurate, i.e. it is corrupted by a high model bias, we want our policy update to rely more on trajectories from the real environment. Contrary, we can use an accurate dynamics model to generate the major part of the update samples from the approximated environment to reduce the sample complexity of PPO. In order to control the amount of trajectories sampled from the real and approximated environment, we utilize a simple indicator for model inaccuracies. If the agent achieves on both the real- and approximated environment about the same total reward (i.e.

$R(\tau) = \sum_{t=1}^{T} r(s_t, a_t))$, it indicates that the dynamics model is accurate in the states which are likely to be explored by the policy. Hence, it is sufficient to sample the major part from the learned dynamics model. We use this notion to adjust the composition of trajectory samples drawn from the real and approximated environment by the following heuristic: The number $\tilde{n}_k$ of trajectories from the learned dynamics model is set to

$$\tilde{n}_k = \max \left( \min \left( \left| \frac{\bar{R}(\tau)}{\bar{R}(\tilde{\tau})} \right|, \left| \frac{\bar{R}(\tilde{\tau})}{\bar{R}(\tau)} \right| \right) \alpha N_r, 1 \right) \tag{16}$$

$$\bar{R}(\tau) = \frac{1}{n_{k-1}} \sum_{i=1}^{n_{k-1}} R(\tau_i) \quad \bar{R}(\tilde{\tau}) = \frac{1}{\tilde{n}_{k-1}} \sum_{i=1}^{\tilde{n}_{k-1}} R(\tilde{\tau}_i)$$

with $k$ being the index of the current PPO iteration. The hyperparameter $\alpha \in [0, 1]$ controls the maximum of trajectories which can be collected from the learned dynamics model. Consequently, the number of trajectories sampled from the real environment is set to

$$n_k = \max(N_r - \tilde{n}_k, n_{min}) \tag{17}$$

where $N_r$ encodes a fixed number of trajectories utilized for a PPO update and $n_{min}$ is the minimal amount of real environment trajectories per PPO update. Algorithm 2 summarizes our approach.

## 4 RESULTS

In this section, we present the evaluation results of our proposed augmented model-free RL algorithms. The experiments aim to compare the performance of our new method shown in Algorithm 2 to state-of-the-art model-free and model-based algorithms. We used the following algorithms:

- **Trust Region Policy Optimization** (Schulman et al. (2015a)): a model-free RL algorithm with constraint policy gradient update
- **Proximal Policy Optimization** (Schulman et al. (2017)): a model-free RL algorithm with clipped 'surrogate' objective
- **Model-Based Meta-Policy-Optimization** (Clavera et al. (2018)): a model-based RL algorithm that uses an ensemble of dynamics model to quickly adapt to new dynamcis model with one policy gradient step

The evaluation process is performed on four simulated OpenAI Gym Benchmarking tasks (Brockman et al. (2016)), i.e., Hopper, Half-cheetah, Swimmer, and Walker, which are based on the the MuJoCo physics simulator (Todorov et al. (2012)). A detailed description of the environments is given in Appendix A.1 and the hyperparameter settings are listed in Appendix A.2. In

---

**Algorithm 2** Model-based Policy Gradient Enrichment

1: **input:** Batch size $N_r$; $n_{min}$; $\alpha$; PPO hyperparameters; dynamics model hyperparameters
2: initialize $\mathcal{D}$ as an empty dataset
3: $n_0 = N_r$
4: $\tilde{n}_0 = 1$
5: **for** $k = 0$ to $N_{iter}$ **do**
6:     initialize $\mathcal{U}_{\mathcal{PPO}}$ as an empty dataset
7:     collect $n_k$ trajectories with the current policy $\pi$ on the real environment and add them to $\mathcal{U}_{\mathcal{PPO}}$ and $\mathcal{D}$
8:     train anchored ensemble of NNs on $\mathcal{D}$ (Algorithm 1)
9:     collect $\tilde{n}_k$ trajectories with the current policy $\pi$ on the approximated environment and add them to $\mathcal{U}_{\mathcal{PPO}}$
10:     compute $\tilde{n}_{k+1}$ with equation 16
11:     compute $n_{k+1}$ with equation 17
12:     run a PPO update step with the data in $\mathcal{U}_{\mathcal{PPO}}$
13: **end for**

---

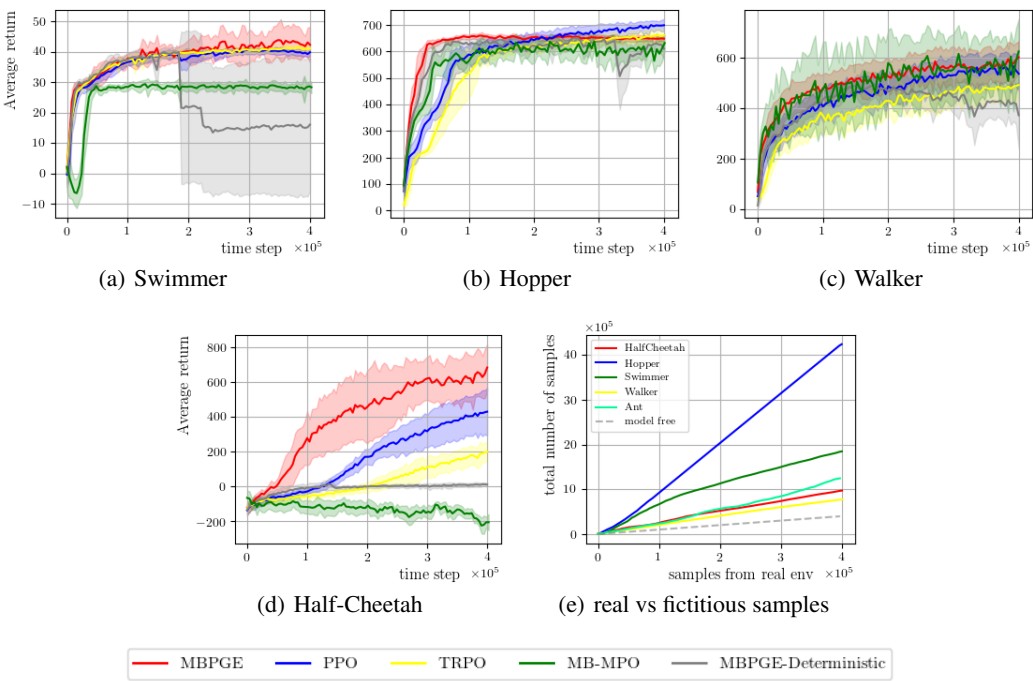

Figure 1: Comparison of learning curves of our approach to other state-of-the-art RL algorithms. The horizontal axis denotes the number of samples, the vertical axis is the average return. The maximum trajectory length is set to 200 time steps. The bottom right figure compares the number samples from the real environment to the total number of used samples for optimization by MBPGE. The result of MBPGE-Deterministic refers to our algorithm in which a deterministic dynamics model was in use, i.e. the next state was not randomly sampled from a predicted distribution but instead chosen to be the point estimate given by the NN.

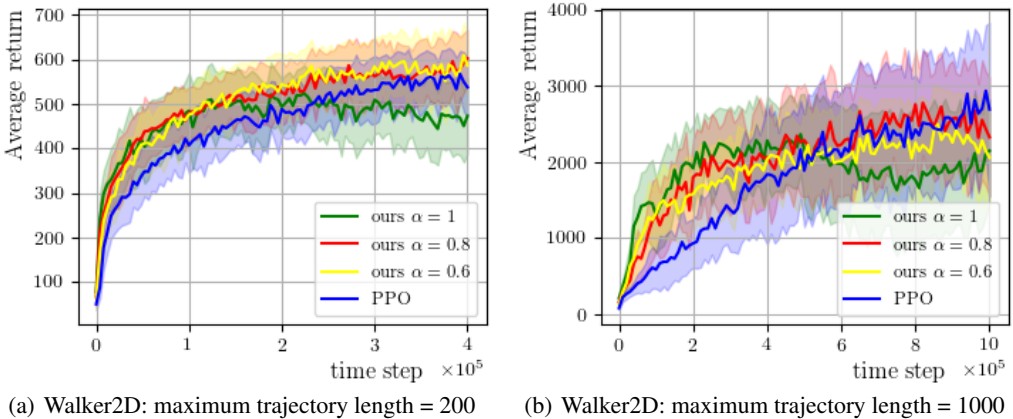

Figure 2: Design study on hyperparameter $\alpha$ in eq.16 to manually restrict the maximum possible number of fictitious data samples. The performance curves are shown for three values, $\alpha = 1$, $\alpha = 0.8$, and $\alpha = 0.6$. The results show that finding the optimal composition of the dataset for a policy gradient estimate is non-trivial.

our experiments we would like to evaluate how our approach compares to state-of-the-art RL algorithms in terms of sample complexity and final performance. The results are performed for a fixed trajectory length of 200 time steps. In total, all algorithms perform $4e5$ steps on the environment. Figure 1 depicts the comparison of our MBPGE algorithm to prior work on pure policy gradient based RL and MB-MPO that exceeds the performance of ME-TRPO as shown in Wang et al. (2019). Furthermore, Figure 1 illustrates the total number of samples (the samples from the real environment plus fictitious samples) our algorithm has processed for optimization.

The results clearly show that MBPGE can outperform the available alternatives. Incorporating fictitious samples from a learned stochastic dynamics can dramatically improve the raw performance of MF-RL. The necessary amount of training samples from the real environment is greatly reduced while the asymptotic performance of MF-RL algorithms is maintained. The positive effect of the stochastic dynamics model becomes apparent by the increased learning rate. As a matter of fact, the averaged return rises significantly faster for MBPGE than for the pure model-free algorithm. We believe that the increased performance is caused by two reasons. First, having no prior on the actual task, the agent needs to explore how to distinguish between high and low-rewarded actions. Since the dataset of the observed experience in an early training stage is simply not accurate enough, the Baysian uncertainty estimate of the randomized anchored MAP in the anchored ensemble will have high variance. Consequently, the expected reward under high variance of the fictitious trajectory samples is very low, even though the mean would be exactly the same as in a deterministic dynamics model. This permits a policy update in the direction of the highest gradient we are certain about while avoiding catastrophic failures due to erroneously optimistic predictions. Second, with progressing data samples the dataset of collected experiences becomes more expressive resulting in a potentially more accurate dynamics model. Figure 1(e) illustrates that the more certain the agent is about its dynamics model the more samples it generates in the fictitious environment. The less challenging the task, the more confident is the agent about its dynamics model and the higher the factor by which the total number of samples used for policy gradient estimates is upscaled (Hopper: 10.62, Swimmer: 4.62, Half-Cheetah: 2.44, Walker: 1.94). Even for the most challenging environment our approach can double the amount of data samples based on the learned dynamics model. As a result, the raw performance of MBPGE is significantly higher.

Designing a suitable heuristic for estimating the best amount of fictitious data samples is very challenging. We tested diverse heuristics with different trajectory lengths and found that the heuristic in eq. 16 performs best (cf. 2). Additionally, we find that the number of fictitious data samples used for a policy update step is, up to a certain degree, of minor importance for learning speed but effects the asymptotic behaviour. As a consequence, we introduced a hyperparameter $\alpha$ in eq. 16 to manually reduce the size of fictitious data samples. Improvement to this inconvenient solution is part of our future research.

## 5 CONCLUSIONS

In this paper, we introduce model-based policy gradient enrichment, an algorithm that incorporates a stochastic dynamics model to augment the dataset for estimating the current policy gradient. Our method takes advantage of randomized anchored MAP that results in an ensemble of neural networks providing a true Baysian uncertainty estimate. We presented that our approach leads to significantly faster learning while maintaining the final asymptotic performance of plain model-free RL. We further show that the stochastic dynamics model is a suitable approach to reduce model bias leading to an increased learning rate in an early training stage.

ACKNOWLEDGMENTS

We thank Maren Bennewitz for helpful discussions and feeback, and Christian Lagemann for comments that greatly improved the manuscript.

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

# Appendices

## A    EXPERIMENTAL DESCRIPTION

### A.1    ENVIRONMENTS

Our method requires the reward of a time step to be a known function of the form $r(s_t, s_{t-1}, a_t)$ or $r(s_t, a_t)$. Otherwise, it would be necessary to introduce another regression model for learning the reward function. Hence, we had to manipulate the default environments slightly as described in the following.

**Hopper:**
The basis of our Hopper was the Hopper-v2 from OpenAI Gym. Its observation vector consists of all angular joint positions and velocities and our only modification was that we extended the observation vector by the $x$ position in space (pseudo code: x = sim.data.qpos[0]). We utilized the default reward function $r(s_t, s_{t-1}, a_t) = \frac{x_t - x_{t-1}}{\Delta t} + 1 - 0.001\|a_t\|_2^2$ and early termination criterion.

**Walker:**
The basis of our Walker was the Walker2d-v2 from OpenAI Gym. Its observation vector consists of all angular joint positions and velocities and our only modification was that we extended the observation vector by the $x$ position in space (pseudo code: x = sim.data.qpos[0]). We utilized the default reward function $r(s_t, s_{t-1}, a_t) = \frac{x_t - x_{t-1}}{\Delta t} + 1 - 0.001\|a_t\|_2^2$ and early termination criterion.

**HalfCheetah:**
The basis of our HalfCheetah was the HalfCheetah-v2 from OpenAI Gym. Its observation vector consists of all angular joint positions and velocities and our only modification was that we extended the observation vector by the $x$ position in space (pseudo code: x = sim.data.qpos[0]). We utilized the default reward function $r(s_t, s_{t-1}, a_t) = \frac{x_t - x_{t-1}}{\Delta t} - 0.1\|a_t\|_2^2$ and no kind of early termination criterion.

**Swimmer:**
The basis of our Swimmer was the Swimmer-v2 from OpenAI Gym. Its observation vector consists of all angular joint positions and velocities and our only modification was that we extended the observation vector by the $x$ position in space (pseudo code: x = sim.data.qpos[0]). We utilized the default reward function $r(s_t, s_{t-1}, a_t) = \frac{x_t - x_{t-1}}{\Delta t} - 0.0001\|a_t\|_2^2$ and no kind of early termination criterion.

**Ant:**
The basis of our Ant was the Ant-v2 from OpenAI Gym. Its observation vector consists of all angular joint positions and velocities as well as some contact related states. Our modification was to remove the contact states from the observation vector and to add the $x$ position in space (pseudo code: x = sim.data.qpos[0]) to it. We utilized the reward function $r(s_t, s_{t-1}, a_t) = \frac{x_t - x_{t-1}}{\Delta t} + 1 - 0.5\|a_t\|_2^2$. The only difference of it to the default reward function is that it has no contact penalty. We additionally utilized the default early termination criterion of Ant-v2.

### A.2    HYPERPARAMETERS

If not specified differently, we used the hyperparameters as listed in Table 1 (MBPGE), Table 2 (PPO), Table 2 (TRPO) and Table 3 (MB-MPO). For our plain PPO and TRPO experiments we have not defined a fixed number of trajectories to collect for a policy update. Instead, we collected trajectories until the batch contained a certain number of time steps $T_{min}$. The Anchored Ensemble requires an assumption about the prior distribution. We have set the mean $\mu_{prior}$ of the Gaussian prior distribution to zero and its variance $\Sigma_{prior}$ according to the Xavier initialization (Glorot & Bengio (2010)) ($\eta_{in} \hat{=}$ fan-in, $\eta_{out} \hat{=}$ fan-out)

$$\Sigma_{prior,i,j} = \frac{2}{\eta_{in,i,j} + \eta_{out,i,j}}$$

within this work. The index $i$ denotes the NN layer and $j$ the unit inside that layer.

| PPO | Anchored Ensemble | MBPGE |
|---|---|---|
| $\lambda_{GAE} = 0.95$ 
 $\gamma = 0.99$ 
 $M = 64$ 
 $\epsilon = 0.2$ 
 Adam stepsize: 0.0003 | $\sigma_\epsilon^2 = 10^{-5}$ 
 $K = 5$ | $N_r = 15$ 
 $n_{min} = 1$ 
 $\alpha = 1$ 
 walker: $\alpha = 0.8$ |

Table 1: MBPGE hyper parameter settings.

| PPO | Trpo |
|---|---|
| $\lambda_{GAE} = 0.95$ 
 $\gamma = 0.99$ 
 $M = 64$ 
 $\epsilon = 0.2$ 
 $T_{min} = 2048$ 
 Adam stepsize: 0.0003 | $\lambda_{GAE} = 0.95$ 
 $\gamma = 0.99$ 
 max KL: 0.01 
 $T_{min} = 2048$ |

Table 2: PPO and TRPO hyper parameter settings.

| TRPO | MB-MPO |
|---|---|
| $\lambda_{GAE} = 0.95$ 
 $\gamma = 0.99$ 
 max KL: 0.01 | inner adaption step size $\alpha = 0.001$ 

 number of meta-optimization steps: 5 

 number of collected trajectories per algorithm iteration: 20 (real environment) 

 40 trajectories were collected from the learned dynamics model for an inner adaption step and for the meta-policy optimization. |

Table 3: MB-MPO hyper parameter settings.

**Dynamics Model:**
The NNs used for representing the dynamics model in MBPGE and in MB-MPO consisted of 4 hidden ReLU layers with 1000, 900, 800 and 700 units (2244000 parameters). (Fu et al., 2015, p. 5) described the positive effect of the second order Markov assumption (i.e. the next state depends on the current- and previous state and action) with regard to the ability of a NN in learning strong nonlinearities, e.g. contacts. Most challenging robotic manipulation or locomotion problems which are typically solved by model free RL include contacts, so in the experiments we used NNs of the form $f(s_t, a_t, s_{t-1}, a_{t-1})$. A problem with this NN/BNN configuration is the very first prediction of a roll out in which $s_{t-1}$ and $a_{t-1}$ are not defined. We handled this problem by using a second NN/BNN $f_0$ which was trained only on the very first roll out transitions $\{s_1, s_0, a_0\}$ (i.e. a dynamic model only for predicting $s_1 = f_0(s_0, a_0)$). $f_0$ consisted of 3 hidden ReLU layers with 50 units each (7268 parameters).

Furthermore, we trained our dynamics models to predict $\Delta s_t = s_{t+1} - s_t$ instead of directly predicting $s_{t+1}$. Additionally, we used the following regularization techniques:

- Early stopping wit GL criterion as specified in (Prechelt, 2012, p. 58)
- Normalization of the training dataset so that it has a zero mean and a unit variance
- Shuffling of the training data
- Weight decay with $\lambda = 0.0001$ (only for MB-MPO)

**Policy:**
For each tested algorithm we utilized a stochastic policy of the form $a_t \sim \mathcal{N}(\mu(s_t), \sigma)$, in which $\mu(s_t)$ was the output of a NN (2 tanh layers with 128 units each) and $\sigma$ was a vector containing the standard deviation of each action. The network's parameters were initialized randomly according to the Xavier initialization (Glorot & Bengio (2010)) and the initial standard deviation vector was set to $\sigma_i = 1 \ \forall \, i \in \{1, \ldots, D_a\}$ with $D_a$ being the dimension of the action vector.

