# OpenReview forum: "AUGMENTED POLICY GRADIENT METHODS FOR EFFICIENT REINFORCEMENT LEARNING"
_ICLR.cc/2020/Conference — Reject_

### Official Review · AnonReviewer1 · 2019-10-22
**Official Blind Review #1**

**Rating:** 1

**Review:**

### Summary

This paper described a model-based RL method which uses learned dynamics models to augment the replay buffer used when training a PPO agent.
Specifically, the agent learns an ensemble of dynamics models, and then performs a PPO updates using a mixture of trajectories sampled from the true environment and trajectories sampled from teh learned dynamics models.
The fictitious trajectories are sampled from different dynamics models in the ensemble, which helps prevent the policy from exploiting the errors of a single model.
The proportion of real/fake trajectories are adapted using the ratio of the real vs. predicted rewards: if the average returns are similar, then the proportions will be approximately equal, but if they are very different then more trajectories from the real environment will be used.

Overall, I think there are some interesting elements to this paper but it is currently not ready for publication. I recommend weak reject for three reasons: limited novelty, the experiments as they stand are not very convincing and the writing needs work.


#### Novelty
The two stated contributions are:
1. the use of principled uncertainty estimates in model-based RL (an anchored ensemble)
2. the routine which adjusts the proportion of real vs. simulated trajectories for learning the policy.

In my opinion, 1. is not much of a contribution since a number of works have already made use of different types of model uncertainty in the context of model-based RL. Here they use a slightly different version (the anchored ensemble, which is similar to the randomized priors used in DQNs) but this is more of a choice of implementation than a contribution in itself (a number of methods for estimating uncertainty exist, such as regular ensembles, dropout masks, BNNs, etc). The manner in which they use the uncertainty is to perform simulations with several different models to avoid overfitting to one, and this was already proposed in the ME-TRPO paper.

The idea of 2. is interesting and in general, adapting between model-based and model-free RL regimes during execution seems to have potential. However the current method seems currently under-explored and heuristic. The fact that it is sensitive to the \alpha hyperparameter (Figure 2) is a bit troubling. Is there a principled way which does not require tuning this hyperparameter? Or reformulating things so that it is less sensitive?


#### Experiments

Although the experiments show some improvement, they are not that convincing. The improvements do not seem statistically significant for Swimmer and Walker, and the asymptotic performance is worse than PPO for Hopper. There is only half-cheetah where the algorithm shows a clear improvment.

Since one of the claimed contributions is the use of the anchored ensemble, there should be an ablation experiment showing this gives an improvement over a standard ensemble, but this is not included.



#### Writing:

One issue with the paper is that it spends *much* too long discussing related work and preliminaries.
The first 5 pages are devoted to this and the proposed method is only introduced on page 6!

Some detailed comments:
- Paragraph 1 in the intro should be drastically cut. The sentence "Recently...challenging tasks" could be followed by references and then a sentence discussing the sample inefficiency and then moving on to the next paragraph. Most readers will be familiar with DQN, actor critic etc.
- Same for paragraphs 2 and 3. A short discussion of the general idea behind model-based RL and its improved sample efficiency, the issue of the policy exploiting model errors, and some references are sufficient.
- Similarly, section 2.1 is mostly unnecessary. It isn't necessary to detail the updates for standard algorithms such as PPO/TRPO unless they are useful for the proposed algorithm.
- Section 2.2 is also too long. They main idea is very simple and is summarized in Equation 13.




**Experience Assessment:**

I have published one or two papers in this area.

**Review Assessment: Checking Correctness Of Derivations And Theory:**

N/A

**Review Assessment: Checking Correctness Of Experiments:**

I assessed the sensibility of the experiments.

**Review Assessment: Thoroughness In Paper Reading:**

I read the paper at least twice and used my best judgement in assessing the paper.

---

### Official Review · AnonReviewer3 · 2019-10-23
**Official Blind Review #3**

**Rating:** 1

**Review:**

### Summary ###

This paper focuses on model based reinforcement learning (RL). Specifically, the authors consider the setting of combining model based and model free RL algorithms by using the learned dynamics model to generate new data for training the model free algorithm. In order to capture the uncertainty of the environment and the model, the author applied Baysian neural network to learn the dynamics of the environment. The authors approximated the true Bayesian inference process by keeping an anchored ensemble of neural networks, where the prior and posterior of network weights are approximately Gaussian. The ensemble of dynamics model is then used to generate data to train a PPO[1] based agent. In order to prevent the agent from exploiting the learned dynamics model, the authors propose a heuristic way of balancing the amount of real data and model generated data by comparing the rewards.

The authors evaluate the proposed algorithm on simulated robotic locomotion environments in MuJoCo, and the results show that the proposed method has better same efficiency compared to baseline methods in some environments.


### Review ###

Overall I think this paper presents an interesting idea in combing model based and model free RL algorithms. The idea is very well presented and authors include empirical evidence to support the proposed method. However I do find a number of shortcomings that need to be addressed.


Pro:

1. The idea for this paper is really well presented. The structure of the paper is well organized and the experiment results are easy to interpret.

2.  The authors provide a detailed description of the configurations and the hyperparameters for each experiments. Such description would be very helpful if the results in this paper are to be reproduced.


Con:

1. I’m not convinced about the magnitude of novelty in this paper. The proposed method seems very similar to ME-TRPO[2] and it seems to me that the novelty comes from the application of Bayesian ensemble techniques and the generated data ratio tuning heuristics. While these variations might be important for the final performance of the proposed method, the paper does not include any ablation study to further justify the importance of these variations.

2. I’m not convinced about some of the performance of some of the baseline methods presented in this paper. In this paper, MB-MPO[3] does not improve at all during training on Half-Cheetah environment. However, in the original MB-MPO paper, the algorithm does improve and the performance seems to be comparable to that of the proposed method in this paper.

3. The experiment results are not very strong for the proposed method. In 3 of 4 environments, the proposed algorithm does not show much advantage over the baseline algorithms. The only environment in which the proposed method shows significant improvement is Half-Cheetah, and I believe that the baseline algorithms might not be properly tuned in this environment.

4. The paper lacks certain baseline comparisons. There are many other model based RL algorithms developed recently, and it would be important to compare to these methods. Some examples would be ME-TRPO[2], SLBO[4] and MBPO[5].


The idea in the paper is well presented and carefully investigated. However, I am still not convinced about the novelty of the proposed idea and the magnitude of performance improvement. Therefore, I would not recommend acceptance before these problems are addressed.



References

[1] Schulman, John, et al. "Proximal policy optimization algorithms." arXiv preprint arXiv:1707.06347 (2017).

[2] Kurutach, Thanard, et al. "Model-ensemble trust-region policy optimization." arXiv preprint arXiv:1802.10592 (2018).

[3] Clavera, Ignasi, et al. "Model-based reinforcement learning via meta-policy optimization." arXiv preprint arXiv:1809.05214 (2018).

[4] Luo, Yuping, et al. "Algorithmic framework for model-based deep reinforcement learning with theoretical guarantees." arXiv preprint arXiv:1807.03858 (2018).

[5] Janner, Michael, et al. "When to Trust Your Model: Model-Based Policy Optimization." arXiv preprint arXiv:1906.08253 (2019).


**Experience Assessment:**

I have published in this field for several years.

**Review Assessment: Checking Correctness Of Derivations And Theory:**

I assessed the sensibility of the derivations and theory.

**Review Assessment: Checking Correctness Of Experiments:**

I carefully checked the experiments.

**Review Assessment: Thoroughness In Paper Reading:**

I read the paper thoroughly.

---

### Official Review · AnonReviewer2 · 2019-10-27
**Official Blind Review #2**

**Rating:** 1

**Review:**

This paper presents a new model-free + model-based algorithm, MBPGE, that trains a policy using a policy gradient algorithm on top of the learned models. Contrary to previous approaches, they use a true Bayesian distribution by means of the randomized anchorized MAP. Furthermore, they combine rollouts from the real environment and from the learned dynamics directly for the policy training, instead of relying just on the ones of learned dynamics, which induces a larger distributional shift.

The paper readability could be improved. First of all, the introduction spans over 2 pages. The readability could be significantly improved by splitting the introduction into introduction (containing the motivation) and related work. Secondly, the preliminary section spans more than their approach, which hints that the original contribution is too incremental. All of the actual method is just combining section 2.1 and 2.2. Lastly, section 3.1 is hard to parse. The authors extend the paper to 9 pages when there was no need for it.

In terms of contribution, it seems that the main contribution is to use the method presented [1] and replace the ensemble with the ensemble of [2] and use PPO [3] instead of TRPO [4]. There is no further insight in the paper, but the fact that the ensemble of [2] works better than the classical ensemble (fact shown in [2]) and that PPO works better than TRPO (fact shown in [3]). The other contribution is to combine samples from the real environment and the from the learned models to train the policy. This however has actually been done in the context of model-based reinforcement learning, see [5]. It would be interesting that the authors compared against different heuristics to choose the ratio between imagined and real rollouts (i.e., constant, annealed, fraction of improvement [1], etc…). Overall, I don’t think that the contributions of this paper are enough for publication.

Regarding the experiment section I strongly believe that the comparison is flawed. The results they report on MB-MPO do not match with the ones in [6, 7]. For instance, half-cheetah does not learn at all while it is an easier environment than Walker and Hopper (and there’s learning in those). Furthermore, given this they should also compare against ME-TRPO and ME-TRPO switching the TRPO with PPO, since this would be a fairer comparison. This section lacks also of a proper ablation analysis to identify how much each element of the proposed algorithm affects the performance the choice of Bayesian ensemble and heuristic to determine the amount of real samples and imagined ones.

At this stage, there is not enough contribution in terms of novelty nor delta in performance.



[1] Thanard Kurutach, Ignasi Clavera, Yan Duan, Aviv Tamar, Pieter Abbeel. Model-Ensemble Trust-Region Policy Optimization.
[2] Tim Pearce, Felix Leibfried, Alexandra Brintrup, Mohamed Zaki, Andy Neely. Uncertainty in Neural Networks: Approximately Bayesian Ensembling.
[3] John Schulman, Filip Wolski, Prafulla Dhariwal, Alec Radford, Oleg Klimov. Proximal Policy Optimization Algorithms.
[4] John Schulman, Sergey Levine, Philipp Moritz, Michael I. Jordan, Pieter Abbeel. Trust Region Policy Optimization
[5] Michael Janner, Justin Fu, Marvin Zhang, Sergey Levine. When to Trust Your Model: Model-Based Policy Optimization
[6] Ignasi Clavera, Jonas Rothfuss, John Schulman, Yasuhiro Fujita, Tamim Asfour, Pieter Abbeel. Model-Based Reinforcement Learning via Meta-Policy Optimization
[7] Tingwu Wang, Xuchan Bao, Ignasi Clavera, Jerrick Hoang, Yeming Wen, Eric Langlois, Shunshi Zhang, Guodong Zhang, Pieter Abbeel, Jimmy Ba. Benchmarking Model-Based Reinforcement Learning.

**Experience Assessment:**

I have published in this field for several years.

**Review Assessment: Checking Correctness Of Derivations And Theory:**

N/A

**Review Assessment: Checking Correctness Of Experiments:**

I carefully checked the experiments.

**Review Assessment: Thoroughness In Paper Reading:**

I read the paper thoroughly.

---

### Decision · Program_Chairs · 2019-12-19

**Decision:**

Reject

**Comment:**

The authors propose a hybrid model-free/model-based policy gradient method that attempts to reduce sample complexity without degrading asymptotic performance. They evaluate their approach on a collection of benchmark tests.

The reviewers raised concerns about limited novelty of the proposed approach and flaws in the evaluation. The authors need to compare to more baselines and ensure that the baseline algorithms are performing as previously reported. Even then, the reported improvements were small.

Given the issues raised by the reviewers, this paper is not ready for publication at ICLR.